# Identities of women who have an autoimmune rheumatic disease [ARD] during pregnancy planning, pregnancy and early parenting: A qualitative study

Denitza Williams[1]*, Bethan Pell[2], Aimee Grant[3], Julia Sanders[4], Ann Taylor[5], Adrian Edwards[1], Ernest Choy[6], Rhiannon Phillips[7]

1 Division of Population Medicine, School of Medicine, College of Biomedical and Life Sciences, Cardiff University, Cardiff, United Kingdom, 2 Centre for the Development and Evaluation of Complex Intervention for Public Health Improvement [DECIPHer], Cardiff University, Cardiff, United Kingdom, 3 Centre for Lactation, Infant Feeding and Translational Research, Swansea University, Swansea, United Kingdom, 4 School of Healthcare Sciences, Cardiff University, Cardiff, United Kingdom, 5 Centre for Medical Education, School of Medicine, Cardiff University, Cardiff, United Kingdom, 6 Division of Infection and Immunity, CREATE Centre, Section of Rheumatology, Cardiff University, Cardiff, United Kingdom, 7 Rhiannon Phillips, Cardiff School of Sport and Health Sciences, Cardiff Metropolitan University, Cardiff, United Kingdom

* WilliamsD74@cardiff.ac.uk

**Data Availability Statement:** The data are all contained within the paper.

## Abstract

### Objective

Women of reproductive age who have autoimmune rheumatic diseases [ARDs] have expressed a need to be better supported with making decisions about pregnancy. Women with ARDs want their motherhood identities and associated preferences to be taken into account in decisions about their healthcare. The aim of this study was to explore the inter-play between illness and motherhood identities of women with ARDs during preconception decision making.

### Methods

Timeline-facilitated qualitative interviews with women diagnosed with an ARD [18–49 years old]. Participants were purposively sampled based on the following three criteria: thinking about getting pregnant, currently pregnant, or had young children. Interviews were themati-cally analysed.

### Results

Twenty-two women were interviewed face-to-face [N = 6] or over the telephone [N = 16]. Interview length ranged from 20 minutes to 70 minutes, with a mean length of 48 minutes. Three main themes were identified: prioritisation, discrepancy, and trade-off. Difficulties in balancing multiple identities in healthcare encounters were reported. Women used 'self-guides' as a reference for priority setting in a dynamic process that shifted as their level of disease activity altered and as their motherhood identity became more or less of a focus at a

**Funding:** This study was funded by a Wellcome Trust Institutional Strategic Support Fund grant (https://wellcome.org/grant-funding/funded-people-and-projects/institutional-strategic-support-fund). The funders had no role in study design, data collection and analysis, decision to publish, or preparation of the manuscript.

**Competing interests:** The authors have declared that no competing interests exist.

given point in time. Women's illness and motherhood identities did not present in isolation but were intertwined.

## Conclusions

Findings highlight the need for holistic person-centred care that supports women with the complex and emotive decisions relating to preconception decision-making. In practice, health professionals need to consider women's multiple and sometimes conflicting identities, and include both their condition and family associated goals and values within healthcare communication.

## Introduction

Women of reproductive age who have autoimmune rheumatic diseases [ARDs] such as rheumatoid arthritis [RA] and systemic lupus erythematosus [SLE] have expressed a need to be better supported with making decisions about pregnancy [1–3]. Many women with ARDs will have positive pregnancy and parenting outcomes, but there are risks involved that need to be considered, including the potential effects of the disease itself and some disease-modifying treatments on fertility, pregnancy and breastfeeding [1, 4–6]. Women with ARDs want their motherhood identities and associated preferences to be taken into account in decisions about their healthcare [2]. In this manuscript, we investigate the development and interplay between illness and motherhood identities of women with ARDs as they navigate the healthcare system.

The diagnosis of chronic illness, such as autoimmune rheumatic disease, can lead to changes in an individual's sense of 'self' as the previous identity is replaced by an 'illness identity' once a diagnosis is made and treatment instigated. [7]. An illness identity includes perceptions in relation to physical changes, emotional reactions to the physical changes, cognitive constructions of the illness and the impact this has on the self-perceived future [8]. When women who have an ARD become mothers, they can be challenged by the symptoms associated with their illness such as fatigue, pain and impaired psychological well-being in addition to the usual challenges presented to women without a long-term condition [1, 9].

The social category of being a mother is not neutral and holds a particular meaning and significance across time and place. There is an ideology of the expectant mother and the identity she must hold [10]. The identity of a 'moral and good' mother is one that is taken on before birth or even pregnancy and embodies the guiding ethic of 'proper' care of children [11]. In contrast to the 'moral and good mother' is that of the 'immoral and bad mother' [10]. The identity of an 'immoral and bad mother' mother would be that of someone who fails to live up to the social ideal of a mother that has been imposed by societal and cultural expectations [10]. The terms 'moral/immoral', 'good/bad' throughout this manuscript are going to be used to reflect the societal judgements placed on individuals and reflected within referenced theories. They are not as a means to stigmatise. Developing the identity of a mother involves internalising perceived ideals of how women ought to function as a mother, whilst understanding the sometimes contradictory lived reality of mothering [12, 13]. Women who do not reach, or struggle to maintain, this ideal identity have been found to feel increasing tension over their shortcomings and have reported negative psychological effects such as feelings of guilt [14, 15].

The Self-Discrepancy Theory [16] suggests that the degree of discrepancy between the actual self [the attributes a person possesses], ought self [who a person feels they should or

must be], and ideal self [who a person hopes or wishes to be] can lead to negative affect, guilt and shame [17]. The values that people attach to their actual, ought, and ideal identities act as 'self-guides' in influencing behaviour and motivation. Analysis of conversations of a UK on-line parenting forum suggested that women can feel pressure to be an unrealistic 'selfless' and 'ideal' mother [18]. Women's perceived shortcomings in fulfilling their own and others' expectations of their role as a mother can leave them feeling dissatisfied and questioning their identity [19, 20]. In the context of women with ARDs, Self-Discrepency Theory would suggest that if mothers experience a discrepancy between their actual self [a woman managing a painful and potentially physically disabling condition], their ideal self [the mother they wish to be] and their ought self [the mother they feel they should be], this would lead to negative self-evaluations [7, 17].

There is a lack of research focusing on exploring the influence of healthcare interactions on women's motherhood and illness identities. Understanding how women's illness and motherhood identities influence, and are influenced by, interactions with healthcare professionals and services is important in designing patient-centred approaches to healthcare. In this study, we use participatory qualitative methods to explore the self-identity of women with ARDs as they consider pregnancy, are pregnant, and/or have young children.

## Methods

### Design

Timeline-facilitated qualitative interviews with women with ARDs who took part in the STAR Family Survey [1].

### Ethical approval

Ethical approval for the study, including the consent process, was granted by the Cardiff University School of Medicine Research Ethics Committee on 20/10/16 REF16/56.

### Participants and recruitment

Women who had expressed an interest in taking part in an interview were sampled from the STAR Family Survey [1]. The STAR Family Survey was a United Kingdom-based cross-sectional on-line survey [n = 128] focusing on women with ARDs' experiences of pre-conception, pregnancy and early parenting.

Interview participants were purposively sampled to broadly achieve equal representation of those who were: thinking about getting pregnant, currently pregnant, or had young children. The aim was to explore the identity in women who were at different stages of starting a family, whilst also managing an ARD. Participants were contacted through e-mail or telephone, based on the contact details provided in the survey. Those who expressed interest in being interviewed were sent a study information pack containing participant information sheet, consent form and stamped return envelope. Participants were provided with a £20 shopping voucher as a thank you for taking part.

### Inclusion criteria

Women aged 18–49 years, who had been diagnosed with an ARD [i.e. inflammatory arthritis or auto-immune connective tissue disease for which people would normally be under the care of a rheumatologist]. Women also needed to be planning on becoming pregnant in the next 5 years, currently pregnant or have a child/children who were under the age of 5 years. Participants had completed the STAR Family Study survey [1] and consented to interview.

### Exclusion criteria

Women who were under 18 or over 50 years of age and those who had a disease not classified as an ARD [e.g. joint hypermobility, fibromyalgia].

### Interview procedure

A flexible narrative approach using visual methods was used to encourage participants to talk about their lived experiences regarding their experiences of identity, motherhood and healthcare in their own words, focusing on things that were important to them, rather than being guided by researcher-generated topics [21]. Before the interviews, women were sent a resource pack containing an example timeline outline, stationary, paper, stickers, and an information sheet outlining examples of the topics we were interested in covering during the interview. The timelines provided a visual tool which enabled women to map out their journey so far. The timelines were used in interviews to provide cues, prompt discussion, provide an opportunity for self-reflection, and help with building a rapport between participants and the interviewer [22, 23]. Preparing a timeline was voluntary and women were encouraged to use their own formats [e.g. notes, diagrams they had drawn] as an alternative if they wished to do so. Interviews were conducted face-to-face or over the telephone. Face-to-face interviews were conducted in the participant's home or at Cardiff University. For more information about the interview materials and procedure please refer to Pell et al [2020] [23]. Interviews were conducted by Denitza Williams, PhD [DW] and Bethan Pell, BSc [BP]. At the time of the interviews, DW was a post-doctoral researcher and BP was a research assistant, both identified as female. Neither interviewer identified as having a long-term condition; DW was a mother to two children, BP did not have any children. The interviewers had no prior relationships with the participants. Informed consent was obtained from participants before the interview [written and verbal].

### Analysis

All interviews were audio-recorded and transcribed verbatim by a professional transcription company. DW conducted the analysis. Data were analysed thematically and analysis was based primarily on social phenomenology [24], focussing on exploring how women made sense of their identity and motherhood choices. The data were analysed using Braun and Clarke's Phases of Thematic Analysis [25] which involved familiarisation, generating initial codes and identifying themes. NViVo10 was used to support the analysis. The protocol did not include double coding of data. Instead, the qualitative research team [DW, BP, AG, RP] held regular data analysis meetings to discuss the themes arising from the data, and the development of the coding framework. Each member of the qualitative research group [RP, AG, DW, BP] contributed their own perspective, relating to disciplinary backgrounds including psychology, sociology, social policy and public health. This approach has been identified as appropriate for enhancing the validity and quality of qualitative research [26]. Analysis was guided by the concept of information power [27] instead of data saturation.

### Results

Twenty-two women were interviewed face-to-face [N = 6] or over the telephone [N = 16]. Participant demographic details are reported in Phillips et al [2018] [28]. Interview length ranged from 20 minutes to 70 minutes, with a mean length of 48 minutes.

Through thematic analysis, three main themes were identified: prioritisation, discrepancy, and trade-off [see Fig 1]. Each of the main themes and sub-themes will be discussed below.

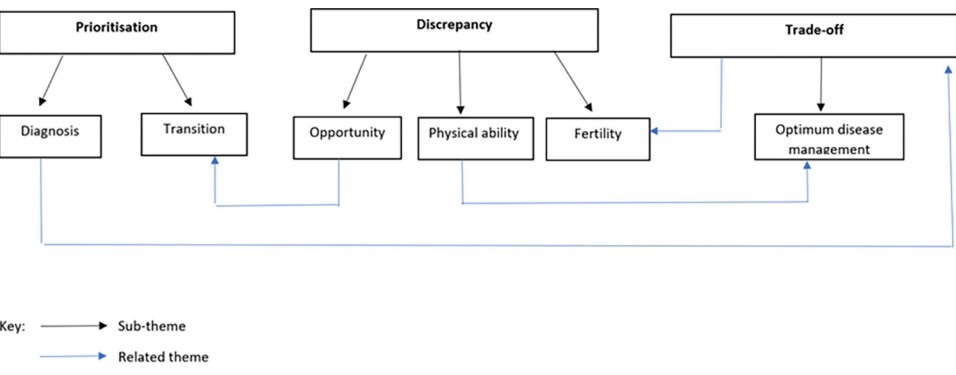

Key: ———▶ Sub-theme
———▶ Related theme

**Fig 1. Thematic map.**

## Prioritisation of motherhood versus illness identities

Women spoke about prioritisation of the disease versus their desire for motherhood. The prioritisation sub-themes of diagnosis and transition were identified. Women were often 'juggling meaningful identities' [29]; their identity as a woman, a mother, and a person with a long term condition.

## Diagnosis: The disease comes first

Women often reflected on the diagnosis of their ARD and their respective treatment options, contextualising their experiences in terms of their identity as an individual who was at crisis point and in desperate need of effective disease management. As disease management was the priority at this point, when the decision to use disease modifying drugs that might have an impact on fertility was made, their future motherhood plans were often not discussed. A woman's own self-identity and the prioritised identify by the clinician at that point was of a sick patient in need of treatment, and not a woman who might in the future like to have a child. Clinical practice guidelines can also contribute to this, as they focus firstly on ensuring optional disease control before the possibility of a pregnancy. This generates a direct impact on the prioritised identity at the time of a consultation.

> *Yeah, I think we did talk about kind of because of the chemotherapy treatments it was likely that possibly I wouldn't be able to conceive. So it was kind of not brushed under the carpet but just one of those ok this has happened and we're talking about it but she's not in the place where she's thinking about having children and actually it's better to be healthy [P1: 37 years old, 1 child, Systemic Lupus]*

## Transition: More than the disease

For women who were diagnosed before becoming a mother, shifts in priority were evident following diagnosis. Women reported that their disease needed to be well-managed to enable them to become a mother. Some women reported concern about the potential effects of their ARD when parenting. Concerns about being able to incorporate children into the 'boundaries of themselves' [30] in terms of the expansion of the self to incorporate children and the associated demands were raised.

> *if we're kind of successful in that [becoming a mother], the impact of having a child to bring up and everything that you need to be able to do with that, am I going to be well enough to do*

*that in terms of sitting and you know being able to provide that child with everything it needs in terms of energy and everything else. [P2: 26 years old, no children undisclosed ARD]*

Women discussed how the decision to have children needed to be planned well ahead of time, because many of the drugs used to treat their ARD have teratogenic effects:

*Yeah I guess with like children it's always been ingrained that it has to be planned and stuff like [P2: 26 years old, no children undisclosed ARD]*

Women needed to transition by modifying the drugs that they were taking to manage their disease before trying for a family. It was felt that this need to plan ahead was forced upon them due to their ARD and felt that most people do not need to actively transition so far in advance, or even to consult a health professional at all when planning to become a mother. The transition signified a motivational and behavioural change between effective management of disease and actively trying to conceive. The women in our study described this transition as a difficult time, because they had made the decision that they would like to have a child but often felt uncertainty about the effect of a pregnancy on their ARD.

*I've got my own kind of own concerns about you know being at a point where I'm well enough to become pregnant, being well during the pregnancy, can I stay well enough for 9 months that I don't need extra medication that would impact on the pregnancy [P9: 39years old, twins,* Inflammatory arthritis*]*

Self-guides are specific representations of the self or others and can include mental representations of valued attributes [31]. Self-guides can originate from diverse sources such as general social norms, particularly if the guides have relevance in the terms of the life experiences of the women [31]. Here, the values attached to women's illness identity and motherhood identity are guiding them in different directions, creating tension as they try to decide which values are most important at a given time.

## Self-discrepancy

Some women identified discrepancies between their perceptions of motherhood ideals and their actual ability to meet these. Sub-themes included discrepancies in fertility, physical ability and opportunity.

**Fertility.**   Women reflected on the impact of their ARD, their perceived fertility and the chances of a pregnancy ending in miscarriage. One woman experienced a miscarriage during a disease flare. She talked about a discrepancy of feelings because she was informed that her poorly managed disease contributed to the miscarriage, she was also aware that women with ARDs are more likely to suffer a miscarriage, but also that miscarriage is common among healthy women. She reported frustration that she had been advised to restart teratogenic medication and to postpone planning for a baby until her disease was fully under control.

*So yeah following the miscarriage I was all for planning to just get pregnant again, but on advice from the rheumatologist and the renal specialist, they you know because I have been so ill that is probably why I miscarried and they have managed to convince me that I'm better off waiting, I'm back on the Micophenolate effectively and I'm in a treatment plan that's not pregnancy friendly to try and get my lupus under control [P15: 32 years of age, no children,* Lupus*]*

This signals a loss of a valued mothering identity, an important part of the self [32]. This signifies a discrepancy between the self-guides held by the health professional and the woman in terms of pursuing a course of action that is consistent with her wish to become a mother [31].

**Physical ability.** The potential impact of physical disability was identified as a factor in motherhood identity. One woman reflected on her social identity in terms of the practical social help she received from a volunteer. The availability of this support helped the mother feel that her children were not missing out on 'normal' childhood due to her long-term condition. This seemed to ease some of the guilt she felt in relation to limited mobility:

*the only thing I have to say that's helped my kids through it all is in [city] there is the young carer's project they are a charitable organisation [. . .] they take kids out every other week to do things like to the cinema or the park, or they go bowling. Once a year they go on a residential trip as well so it's kind of to give them a break from caring [P14: 34years of age, 3 children, Lupus]*

Women reported that their physical disabilities affected their ability to participate in social group activities relating to motherhood. For example, most women had attended a local mother and baby group, but found it difficult to participate, mainly because activities were often floor based. Women reported feeling embarrassed that they were unable to participate like other mothers, questioning their motherhood identity and highlighting the discrepancy between their perceived ought and actual self. For example, one woman took her child to a baby massage group. The activity was floor based and due to the effect of the ARD on her physical abilities she struggled to get back up from the floor and to pick her baby up:

*If I was on the floor I couldn't get up so the first couple of weeks I went I was sat on a chair leaning over [daughter] but then I just it started to make my back ache and everything else and I think for like I wasn't quite bonding with her because it felt like she was so far away so I thought sod it I'll get on the floor with her but that would mean that I would have to crawl to the chair–hand on to something to get myself up and I would literally have to haul myself up off the floor, but yeah I needed somebody to pick <<daughter1>> up off the floor for me and either carry her to the car, or put her in the sling that I had [. . .] you feel like the odd [one] [P4: 40 years old, 1 child, Dermatamyositis]*

This event led to a sense of shame and isolation. Motherhood can have an effect on women; their positive and negative qualities can be heightened when reflecting on the mother-child relationship [28].

*the biggest thing for me is the kids they miss out on so much because of this stupid thing with me. I feel rotten because they have to be no holidays forever and all their pals are going to Spain, or Italy or Portugal or States or holidays and my kids are saying maybe we should go to Norway or Finland because it's so cold and it's not that sunny [it wouldn't cause disease flare] [[crying]] [P14: 34years old, 3 children, Lupus]*

Women felt that they received discrepancies in recommendation from health professionals, especially for breastfeeding. They were often encouraged to initiate and maintain breastfeeding by midwives and health visitors, even though felt it was not the best thing for themselves or their child at that time due to their condition:

*there is such a drive to push you to breastfeed as it is best for the baby and I accepted it, but I also accept that sometimes the best thing in a research report isn't the best thing for the individual and in our case me and my two little ones it certainly wasn't and I think that was why in terms of the health visitor and the midwife and the breastfeeding advocates they're so hell bent on getting a twin mum to breastfeed, because it's that extra pressure again that they focus so much on that they forget the rest of the situation [P8, 37years old, 1 child, Idiopathic Juvenile Arthritis]*

Although participants reported that there was a general pressure to continue breastfeeding, some mothers felt that they were pressured to stop breastfeeding by family members or health professionals, so that they could restart required certain disease modifying medications. Many women reported feeling guilty about their inability to breastfeed, or if they had to stop breast-feeding to re-start medication. Reports of feeling guilty seemed to stem in part from healthcare professional and peer attitudes towards formula feeding. Women reported feeling guilty and ashamed of their desire not to breastfeed, inability to breastfeed or their decision to stop breastfeeding:

*"The breastfeeding advocates had been trying to get me to consider alternative medication and stuff which in an ideal world when we have lots of time maybe that would be a nice thing to do, but when things go downhill drastically you know and I was in a lot of pain and I was struggling to hold 2 babies to breastfeed as well because my joints were sore. . . I marched on and then at 6 weeks I dropped a child. . . it took me actually dropping the child I think that made me think she was worth listening to" [P8, 37years old, 1 child, Idiopathic Juvenile Arthritis]*

**Opportunity.** One woman reflected on the impact of her physical disability on her actual self as a mother. Her baby had an accident at home which needed hospital treatment. Due to her lack of mobility, the woman could not drive to take her child to the hospital independently and had to rely on a neighbour for help. The woman felt that she did not have the opportunity to behave as a 'normal' mother who could react fast and take immediate action to help her child. This intensified her feelings of self-discrepancy and guilt.

*"[The] gap underneath the bathroom door was quite big, he [son] must've put his toes underneath it and either pulled the door, or pushed it so he chopped off the end of his big toe it was kind of hanging you can see the bone it was so bad. And I was in my jammies, I was so tired. I asked my neighbour to get my other 2 boys ready for school and get them to school in time and I just took a taxi [. . .] because I can't drive [. . .] I was again feeling totally useless because I should have been able to drive him, I should've been able to just get dressed get him in the car and to the hospital and I couldn't put one foot in front of the other I was so tired [. . .] it was horrible you know that feeling that, that you should be able to do more for your kids" [P14: 34years old, 3 children, Lupus]*

A few women talked about the possibility of adopting a child but felt that other people's perceptions of them due to their ARD limited their opportunities. One woman and her husband felt that managing her ARD, as well as having a successful pregnancy, would be too difficult as altering her medication was likely to have a detrimental impact on her health. She felt that her ability to be a mother was pre-determined by adoption agencies who made judgements in relation to her long-term condition.

*I would have to prove my ability to be a parent so they wouldn't take [stepchildren] at face value, my step children and I would have to go to parenting classes, they'd have to see evidence of me walking, carrying, being able to, they wanted me to have a–you know a pretend baby at home that they can download and things to prove that I was able to meet its needs all throughout the day and night endlessly and it was really full on [husband] and I got furious and I, I was really pissed off at them I remember saying to them you know in this day and age where you can't discriminate against colour, or sexuality you feel you can discriminate against disability you know [P8, 37years old, 1 child, Idiopathic Juvenile Arthritis]*

Here, there appears to be a discrepancy between women's actual and ideal self-identity and agencies' perceptions of what a mother ought to be.

## Trade-offs between competing values

The complex identity changes women go through when planning to become a mother can require multiple types of trade-offs, including disease management, mothering aspirations and mothering ideals. If women's identities have developed in ways in which they are committed to becoming a parent and being part of a family, the threat their illness poses to their ability to do this means they have to revisit that commitment, which can lead to uncertainty and psychological distress. At this stage, women needed to reflect on their goals and values and what the future holds for them as well as their ability to fulfil their perceived identity of a 'good' mother [10]. This can sometimes be influenced by perceived or experienced ableist societal judgments placed upon women who at that time do not possess the socially described 'essential' criteria for motherhood [33, 34].

Women reflected on their pre-conception journey and the transitions that needed to be made in terms of their identity as a person with a disease that needed to be managed and as a woman who wanted to become a mother. It was often felt that women needed to balance a trade-off between the self as a patient in need of medication to control their ARD [the actual self] and the self as an individual who is well enough to be a mother to meet their own expectations and that of society about mothering [the ought self]. However, the 'ideal' self is also present here in the sense that women's own goals and aspirations, which might be at odds with the 'ought self', that is what they think others expect of them.

Planning a family involved a balance of clinician advice about the trade-offs associated with the management of an ARD and maternal desire. They had to balance multiple identities and expected roles, that of a woman who has a long-term condition and that of a woman who wants to have a child or to expand her family. Some women felt that their motherhood identity was being threatened by the management of their ARD and healthcare communication:

*So they then decided they wanted to start me on floudamide I think it's called which was a disease modifying thing but they told me I couldn't pregnant on it and that the likelihood of me taking it would affect my fertility. So at that point we said well hang on a minute you know I'm only sort of 25, 26, don't really want to start taking stuff that going to affect my fertility, maybe we should just put up with stuff the way it is, try and get pregnant, have another baby because we always knew we wanted a second child then go on the drugs at that point if its needs it. [P3, 34years old, 2 children, Vasculitis]*

Others spoke of specific trade-offs in relation to the management of their disease and their desire to have children. One woman reflected on how she had traded optimum disease management with the chance of conceiving a baby despite an active disease flare. For her, the

priority at that time was to try to ensure that she had a chance of having a child, and her clinical team seemed to be supportive by putting in place necessary referrals:

> *things were flaring back up and they said well look, what we can do is we can do a referral to the fertility clinic, they can take a look and see if there's anything specifically wrong that's stopping it you know you may be eligible for IVF or you know with a discussion with them it might be that they're able to take your eggs and freeze them so that we can start the medication and do it that way. [P3, 34 years old, 2 children, Vasculitis]*

Women with ARDs who had successful pregnancies felt that they were "lucky" because they identified themselves at a higher risk of complications on their journey to motherhood compared to women who do not have a long-term condition. They spoke of trade-offs in relation to miscarriage risk and the chance of a successful pregnancy:

> *I am very very grateful for my two children because there's a lot of people who really suffer and suffer many miscarriages and stuff whereas I have only suffered one miscarriage in my life, a lot of people have suffered a lot worse and haven't been able to have children or have had their fertility robbed because of the medications they've been on and stuff. [P3, 34 years old, 2 children, Vasculitis]*

## Discussion

In this study, we set out to investigate the development and interplay of women's self-identity as a person with an ARD and as a mother as they navigated the healthcare system during preconception planning, pregnancy, and early parenting. Prioritisation of illness versus motherhood identities, self-discrepancy, and trade-offs between competing values were identified as major themes in the discussion surrounding women's experiences. The difficulties in balancing multiple identities in healthcare encounters was evident, both for the women and for the healthcare professionals. Women seemed to use 'self-guides' [31] as a reference for priority setting in a dynamic process that shifted as their level of disease activity altered and as their motherhood identity became more or less of a focus at a given point in time. Women often felt that they needed to balance their 'actual' self [individual with a long-term condition] with their perceptions of the 'ought' self [individual well enough to meet their own as well as society's expectations of being a mother] as well as perceptions of the 'ideal self' [women's own goals and aspirations]. Self-guides became at odds with each other when the action that was needed to manage disease well conflicted with the action required to become a mother or to be a 'good mother' as perceived by the individual. Motherhood identities were also sometimes in conflict with medical recommendations in relation to achieving optimum disease management. When these discrepancies in identity and associated self-guides came into conflict [16], women reported feelings of anxiety, shame, guilt, and frustration. Our findings highlighted the need to consider women's multiple and sometimes conflicting identities, with their associated goals and values, in providing patient-centred healthcare. It is possible that clinical guidelines and recommendations for optimum disease can have a direct impact on identities felt my women and it is therefore important that women's goals and identities as an individual are explored. A person-centred care approach can help facilitate this.

It has been argued that a mother's pregnancy and long-term condition must be viewed as separate but co-existing entities [35]. This suggests that women view their identities separately, with the identity as a mother working separately to the identity as someone who has a long-

term illness. Women's illness and motherhood identities did not present in isolation in our study, but rather were intertwined, highlighting the need for holistic patient-centred care that supports women with the complex and emotive decisions that they need to make. The need for holistic healthcare is reflected in the ethos of the socio-ecological model of health [36] which understands health to be defined more broadly by physical as well as mental and social wellbeing. The socio-ecological model defines health as a broader concept than the active disease status of a person. It can be argued that women's individual and social goals including motherhood identity and reproductive decision making are all part of the overall 'health' of the individual and should be viewed holistically.

Previous research has found that the accommodation and recognition of identity shifts in healthcare communication are imperative for supportive patient decision-making [37]. A patient-centred communication style can facilitate such discussions in consultations, placing 'what matters most to the patient' first. Person-centred care encompasses 'Shared decision-making' a process where health professionals provide understandable information, discuss the pros and cons associated with different treatment options, and fully involve patients in treatment decisions, considering their personal circumstances and preferences [38, 39]. This enables patients to make more informed decisions that align with their preferences and thus fit better with their identity, become more active and empowered in their own healthcare, to have better relationships with their health care professionals, and to feel more satisfied with the choices that they make [40]. There is strong policy support for the routine use of shared decision-making [SDM] in clinical practice [41, 42]. The National Institute for Health and Care Excellence recently published guidance for the incorporation of SDM within healthcare [43]. To facilitate a person-centred approach to pre-conception decision making for women with an ARD, changes at individual and organisational level are needed [29] to create supportive and holistic health care policies. Recommendations from this study include: investment in shared decision-making training for clinical staff, incorporation of a broader focus on reproductive and pre-conception health options in consultations and raised awareness of how conversations can impact on women's identities and reproductive goals. Adopting SDM in healthcare teams would enable both illness and motherhood identities and associated goals and values to be considered by women and their healthcare teams so that women's preferences and priorities can be better incorporated in healthcare decisions.

## Strengths and limitations

This study explored women's perceived illness and motherhood identities using in-depth participant centred interviews. This enabled us to understand the way in which women perceived their motherhood and illness identities and the way this influenced motivation and behaviour in relation to the management of disease and becoming [or being] a mother. The women who took part in our interviews were more highly educated and more likely to be employed than the overall STAR Family survey participants, which is indicative of a sampling bias that should be taken into consideration when generalising from the findings. Furthermore, this study was conducted before the global COVID-19 pandemic, which saw significant shifts in the delivery of healthcare, with modifications to services including remote consultations for non-urgent healthcare appointments, and significant changes in the provision of maternity services [44]. Therefore, findings might not be applicable to all women with ARDs or other long-term conditions. Further research should explore experiences in those under-represented in this study and the changes during the COVID pandemic which might have impacted on healthcare communication.

## Conclusion

This study explored self-identity during pre-conception, pregnancy and early parenting in women with an autoimmune rheumatic disease related to healthcare communication and decision-making. Women's illness and motherhood aspirations were key parts of the formation of their self-identity. Women were balancing and shifting through intertwined illness and mothering identities, which sometimes came into direct conflict with each other. It is imperative that both illness and motherhood identities are acknowledged and explored during healthcare encounters to facilitate preference-sensitive healthcare decisions. A person-centred approach to healthcare communication would facilitate this by encouraging dialogue between healthcare professionals and women about preferences associated with treatment decision-making that incorporate both their illness and mothering identities. Additional research will be required to develop and test the feasibility and acceptability of a person-centred intervention to be used in healthcare encounters.

## Acknowledgments

The authors wish to thank all of the women that took part in this research. We would like to thank our patient involvement representatives for their continued support with this work.

## Author Contributions

**Conceptualization:** Aimee Grant, Julia Sanders, Ann Taylor, Adrian Edwards, Ernest Choy, Rhiannon Phillips.

**Formal analysis:** Denitza Williams.

**Funding acquisition:** Julia Sanders, Ernest Choy, Rhiannon Phillips.

**Investigation:** Denitza Williams.

**Methodology:** Denitza Williams, Bethan Pell, Aimee Grant, Julia Sanders, Ann Taylor, Ernest Choy, Rhiannon Phillips.

**Project administration:** Rhiannon Phillips.

**Supervision:** Ann Taylor, Adrian Edwards, Ernest Choy, Rhiannon Phillips.

**Writing – original draft:** Denitza Williams, Rhiannon Phillips.

**Writing – review & editing:** Bethan Pell, Aimee Grant, Julia Sanders, Ann Taylor, Adrian Edwards, Ernest Choy.

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
