## [Decision Letter · Decision Letter 0]

24 Aug 2022

PONE-D-22-02786Identities of women who have an autoimmune rheumatic disease (ARD) during pregnancy planning, pregnancy and early parenting: A qualitative studyPLOS ONE

Dear Dr. Williams,

Thank you for submitting your manuscript to PLOS ONE. After careful consideration, we feel that it has merit but does not fully meet PLOS ONE’s publication criteria as it currently stands. Therefore, we invite you to submit a revised version of the manuscript that addresses the points raised during the review process.

The reviewers have raised a number of major concerns. They feel the manuscript should outline a clearly-defined research question, and they request improvements to the reporting of methodological aspects of the study.

Could you please carefully revise the manuscript to address all comments raised?

We look forward to receiving your revised manuscript.

Kind regards,

Thomas Phillips, PhD

Staff Editor

PLOS ONE

Journal Requirements:

Reviewers' comments:

Reviewer's Responses to Questions

**Comments to the Author**

1. Is the manuscript technically sound, and do the data support the conclusions?

Reviewer #1: Yes

Reviewer #2: Yes

2. Has the statistical analysis been performed appropriately and rigorously? 

Reviewer #1: N/A

Reviewer #2: Yes

3. Have the authors made all data underlying the findings in their manuscript fully available?

Reviewer #1: No

Reviewer #2: Yes

4. Is the manuscript presented in an intelligible fashion and written in standard English?

Reviewer #1: Yes

Reviewer #2: Yes

5. Review Comments to the Author

Reviewer #1: I found this to be a very interesting paper that explores a lot of internal conflict experienced by women living with autoimmune rheumatic diseases. Although the topic of information needs and priorities of this population have been studied (and reproduced), I am not aware of any research on this particular research question and the emotional impact on patients and their families. I would ask that any results of literature reviews be clearly noted (briefly) in the introduction. I also offer the following suggestions and feedback for consideration by the study team:

- I'm not sure I'm entirely comfortable with the "moral and good mother" reference. Although I think I understand what the writer is trying to get across, I wonder if there is a less stigmatizing word that could be used in its place. I believe the authors are trying to convey the norms, standards, etc. imposed by society on women which is indeed correct, but wonder if the current wording gets this idea across effectively to a perhaps less informed reader.

- Did the study team consider how disability and systemic ableism might impact on often conflicting identities? There is a specific mention of physical disability later in the paper and I wonder if there can be a sentence or two of reflection about this phenomenon. I do not think that society's views of disability support women with ARD's in choosing motherhood or in being mothers. Perhaps elements of the socio-ecological approach could be helpful in this regard. I'd recommend a sentence of reflection in the discussion section. To me, the social vs medical model of disability has significant impacts on women with ARD's making reproductive health decisions.

- I quite liked the flexible approach taken with the use of visual elements and 'resource pack' to help women share their experiences. Might it be possible to include what this looked like in an appendix? It might be of value to readers doing similar research.

- I would remove "lived experiences" in quotations (line 144). It is merely a fact and way of explaining the keepers of the knowledge being sought.

- I also really like how you all reflected on the personal roles and perspectives you took in collecting the data and that guided the analysis.

- In the results section, I wonder if you might briefly explain how others shaped the identities taken on by women in a more direct way. This relates to how women are affected by societal norms and values including those of health care providers and whatever guidelines are in place (e.g. current recommendations - see ACR and EULAR that disease be well controlled). I'm not sure if researchers and health care providers necessarily see the unexpected consequences of their recommendations and clinical practice guidelines.

- I wonder if it might be valuable to reflect on the role of policy makers in creating supportive policies, health care advice, and built environments that support mothers with ARD's including accessible programs. Perhaps a nice addition to the discussion section.

- I really like the addition of SDM and patient preferences in the discussion section.

Reviewer #2: The article is very relevant and interesting. Pregnancy in autoimmune rheumatic diseases is challenging and a person-centered approach is of paramount importance. This study helps identify ideas, themes, and difficulties persons have during their disease and their pregnancy or pregnancy planning.

Qualitative study well carried out, good methodology, good writing and with value in its findings

Consider adding the diagnoses of the ARDs that are included and the activity of the disease at the time of the interview. Include sociodemographic characteristics.

The results may not be applicable to other populations.

6. PLOS authors have the option to publish the peer review history of their article (what does this mean?). If published, this will include your full peer review and any attached files.

Reviewer #1: **Yes: **Laurie Proulx

Reviewer #2: **Yes: **Dionicio A. Galarza-Delgado, MD PhD

---

## [Author Response · Author response to Decision Letter 0]

11 Oct 2022

05.10.2022

Dear Dr Phillips,

We would like to thank the reviewers for taking the time to review the manuscript and for their insightful comments and feedback. 

We have amended the manuscript accordingly and have responded to each comment below in blue. Where modifications have been made to the manuscript we have copied the modified text (in italics) within this response and also included the corresponding line number within the manuscript for ease of review.

We hope that the comments and modifications outlined below are satisfactory for recommending publication. We have also ensured that the clean ‘manuscript’ adheres to PLOS ONE's style requirements.

I look forward to hearing from you.

Kind Regards

Dr Denitza Williams

Reviewer #1: I found this to be a very interesting paper that explores a lot of internal conflict experienced by women living with autoimmune rheumatic diseases. Although the topic of information needs and priorities of this population have been studied (and reproduced), I am not aware of any research on this particular research question and the emotional impact on patients and their families. I would ask that any results of literature reviews be clearly noted (briefly) in the introduction. 

We would like to thank the reviewer for their feedback. We acknowledge that there is a lack of research on this specific topic within the literature. We have made this more explicit in the introduction of the manuscript.

“There is a lack of research focusing on exploring the influence of healthcare interactions on women’s motherhood and illness identities.” (Lines 110-111)

 I also offer the following suggestions and feedback for consideration by the study team:

- I'm not sure I'm entirely comfortable with the "moral and good mother" reference. Although I think I understand what the writer is trying to get across, I wonder if there is a less stigmatizing word that could be used in its place. I believe the authors are trying to convey the norms, standards, etc. imposed by society on women which is indeed correct, but wonder if the current wording gets this idea across effectively to a perhaps less informed reader. 

We would like to thank the reviewer for their feedback and consideration of the sensitive nature associated with using the terms ‘moral’ and ‘good’. We agree that they might be perceived as potentially stigmatising to a reader less familiar with the theories used in the manuscript and appreciate the reviewer highlighting this. We have therefore included a definition of the terms outlining that they are not used as a means to stigmatise but as a way of signposting society’s norms and beliefs and reflecting the terms used in the theories referenced.

“The terms ‘moral/immoral’, ‘good/bad’ throughout this manuscript are going to be used to reflect the societal judgements placed on individuals and reflected within referenced theories. They are not as a means to stigmatise. (Lines 89-91)”

- Did the study team consider how disability and systemic ableism might impact on often conflicting identities? There is a specific mention of physical disability later in the paper and I wonder if there can be a sentence or two of reflection about this phenomenon. I do not think that society's views of disability support women with ARD's in choosing motherhood or in being mothers. Perhaps elements of the socio-ecological approach could be helpful in this regard. I'd recommend a sentence of reflection in the discussion section. To me, the social vs medical model of disability has significant impacts on women with ARD's making reproductive health decisions.

Thank you for your feedback. The study did not seek to specifically explore how disability and systemic ableism impact on conflicting identities, however as noted we did discuss findings in relation to women’s perceived identity and their potential disabilities. We have added a sentence reflecting on the impact of perceived/experienced ableism in the results section specifically in the ‘trade-offs between competing values’ results section.

“This can sometimes be influenced by perceived or experienced ableist societal judgments placed upon women who at that time do not possess the socially described ‘essential’ criteria for motherhood (34, 35).” (Lines 376-378)

We agree that the social ecological model for health is a suitable way of contextualising the definition of ‘health’ within the broader sense and is useful for this paper. We did not seek to apply the model to the study, however we have reflected about the model in the discussion section. 

“The need for holistic healthcare is reflected in the ethos of the socio-ecological model of health (34) which understands health to be defined more broadly by both physical and mental and social wellbeing. The socio-ecological model defines health as a broader concept than the active disease status of a person. It can be argued that women’s individual and social goals including motherhood identity and reproductive decision making are all part of the overall ‘health’ of the individual and should be viewed holistically.” (452-457)

- I quite liked the flexible approach taken with the use of visual elements and 'resource pack' to help women share their experiences. Might it be possible to include what this looked like in an appendix? It might be of value to readers doing similar research.

Thank you. The approach which was used for this study is reported in our paper led by Pell at al 2020. We will direct readers to this publication for more information about the materials that were used. The paper outlines the use of the qualitative approach both over the telephone and face-to-face and provides copies of the participant materials used. 

The paper we refer to is: Pell, B., Williams, D., Phillips, R., Sanders, J., Edwards, A., Choy, E. and Grant, A. 2020. Using visual timelines in telephone interviews: Reflections and lessons learned from the STAR Family Study. International Journal of Qualitative Methods 19, pp. 1-11. (10.1177/1609406920913675)

The following text has been added in the Methods section to orientate the reader: 

“For more information about the interview materials and procedure please refer to Pell et al (2020) (24).” (Line 160-161)

- I would remove "lived experiences" in quotations (line 144). It is merely a fact and way of explaining the keepers of the knowledge being sought.

Thank you for raising this point, we have removed the quotations from the words (Line 148).

- I also really like how you all reflected on the personal roles and perspectives you took in collecting the data and that guided the analysis.

Thank you very much for this feedback. The team were very conscious to ensure that we were reflective in the process.

- In the results section, I wonder if you might briefly explain how others shaped the identities taken on by women in a more direct way. This relates to how women are affected by societal norms and values including those of health care providers and whatever guidelines are in place (e.g. current recommendations - see ACR and EULAR that disease be well controlled). I'm not sure if researchers and health care providers necessarily see the unexpected consequences of their recommendations and clinical practice guidelines.

Thank you for this consideration. We have included a sentence to orientate the reader to this issue within the ‘Diagnosis: disease comes first’ section of the results:

“Clinical practice guidelines can also contribute to this, as they focus firstly on ensuring optional disease control before the possibility of a pregnancy. This generates a direct impact on the prioritised identity at the time of a consultation.” (Lines 204-206).

We have also reflected on this point in the discussion section of the article:

“It is possible that clinical guidelines and recommendations for optimum disease can have a direct impact on identities felt my women and it is therefore important that women’s goals and identities as an individual are explored. A person-centred care approach can help facilitate this.” (Lines 442-445)

- I wonder if it might be valuable to reflect on the role of policy makers in creating supportive policies, health care advice, and built environments that support mothers with ARD's including accessible programs. Perhaps a nice addition to the discussion section.

Thank you for raising this point. The need for incorporation of a shared decision making approach within healthcare was discussed in the discussion section including the policy support from the National Institute for Health and Care Excellence guidance for the incorporation of SDM within healthcare. However, we have further reflected on the role of individual and organisational level guidance and change needed to create supportive and holistic healthcare policies. 

“To facilitate a person-centred approach to pre-conception decision making for women with an ARD, changes at individual and organisational level are needed (30) to create supportive and holistic health care policies. Recommendations from this study include: investment in shared decision-making training for clinical staff, incorporation of a broader focus on reproductive and pre-conception health options in consultations and raised awareness of how conversations can impact on women’s identities and reproductive goals.” (Lines 471-477)

- I really like the addition of SDM and patient preferences in the discussion section.

Thank you for this feedback.

Reviewer #2: The article is very relevant and interesting. Pregnancy in autoimmune rheumatic diseases is challenging and a person-centered approach is of paramount importance. This study helps identify ideas, themes, and difficulties persons have during their disease and their pregnancy or pregnancy planning.

Qualitative study well carried out, good methodology, good writing and with value in its findings

We would like to thank the reviewer for their feedback.

Consider adding the diagnoses of the ARDs that are included and the activity of the disease at the time of the interview. Include sociodemographic characteristics.

We would like to thank the reviewer for this comment and agree that including some characteristics related to the quotes presented will be useful in orientating the reader. Demographic characteristics for our sample are reported in our previously published paper by Phillips et al 2018. We will orientate the reader to this in the results section.

“Participant demographic details are reported in Phillips et al (2018)(29).” (Line 182)

We have added description to each of the participant quotes relating to the age of the participant, whether they have any children and how many and also their disease.

For example: “(P1: 37 years old, 1 child, Systemic Lupus)”

The results may not be applicable to other populations.

We would like to thank the reviewer for this comment. We agree that findings might not be applicable to other populations in terms of demographic and disease. We have therefore expanded the end of the limitations section to further clarify this (new addition in blue text).

“The women who took part in our interviews were more highly educated and more likely to be employed than the overall STAR Family survey participants, which is indicative of a sampling bias that should be taken into consideration when generalising from the findings. Furthermore, this study was conducted before the global COVID-19 pandemic, which saw significant shifts in the delivery of healthcare, with modifications to services including remote consultations for non-urgent healthcare appointments, and significant changes in the provision of maternity services (42). “Therefore, findings might not be applicable to all women with ARDs or other long-term conditions. Further research should explore experiences in those under-represented in this study and the changes during the COVID pandemic which might have impacted on healthcare communication.” (Lines 402-495)

---

## [Editor Report · Decision Letter 1]

21 Oct 2022

Identities of women who have an autoimmune rheumatic disease (ARD) during pregnancy planning, pregnancy and early parenting: A qualitative study

PONE-D-22-02786R1

Dear Dr. Williams,

We’re pleased to inform you that your manuscript has been judged scientifically suitable for publication and will be formally accepted for publication once it meets all outstanding technical requirements.

Kind regards,

Esmat Mehrabi

Academic Editor

PLOS ONE